# Effects of a Fruit and Vegetable-Based Nutraceutical on Biomarkers of Inflammation and Oxidative Status in the Plasma of a Healthy Population: A Placebo-Controlled, Double-Blind, and Randomized Clinical Trial

**DOI:** 10.3390/molecules26123604

**Published:** 2021-06-12

**Authors:** Raúl Arcusa, Juan Ángel Carrillo, Raquel Xandri-Martínez, Begoña Cerdá, Débora Villaño, Javier Marhuenda, María Pilar Zafrilla

**Affiliations:** Faculty of Health Sciences, Universidad Católica de San Antonio, 30107 Murcia, Spain; rarcusa@ucam.edu (R.A.); jacarrillo4@alu.ucam.edu (J.Á.C.); rxandri@ucam.edu (R.X.-M.); bcerda@ucam.edu (B.C.); dvillano@ucam.edu (D.V.); mpzafrilla@ucam.edu (M.P.Z.)

**Keywords:** polyphenol, biomarkers, oxidative stress, fruit, vegetable, nutraceutical

## Abstract

There is scientific evidence of the positive effect of polyphenols from plant foods on inflammation and oxidative status. The aim of the present study was to investigate whether treatment with a high-polyphenolic nutraceutical reduces the plasmatic concentration of certain oxidative and inflammatory biomarkers in a healthy population. One hundred and eight subjects were selected and stratified by sex in the intervention group (*n* = 53) and the placebo group (*n* = 55). Ninety-two subjects completed the study after two 16-week treatment periods separated by a four-week washout period. The results revealed statistically significant differences in subjects treated with the polyphenolic extract compared to the placebo: A decrease in homocysteine, oxidized low-density lipoprotein (OxLDL), TNF-α, sTNFR1, and C-reactive protein (CRP). The most significant decrease was observed for OxLDL (from 78.98 ± 24.48 to 69.52 ± 15.64; *p* < 0.05) and CRP (from 1.50 ± 0.33 to 1.39 ± 0.37; *p* < 0.05), both showing significant differences compared to the placebo (*p* < 0.001). Moreover, catecholamines increased after the administration of the product under investigation, especially in the case of dopamine (from 15.43 ± 2.66 to 19.61 ± 5.73; *p* < 0.05). Therefore, the consumption of a nutraceutical based on fruit and vegetables with a high polyphenol content seems to improve the parameters related to health benefits (oxidative and inflammatory biomarkers), including remarkable changes in the expression of catecholamines.

## 1. Introduction

The scientific literature has documented the role of diet in the prevention of non-communicable diseases such as obesity, cardiovascular diseases, cancer, and all causes of death, especially through the consumption of fruits and vegetables for their high content of bioactive compounds, mainly polyphenols [1,2]. There is an inverse relationship between the Mediterranean diet, characterized by a high consumption of products rich in polyphenols, through the consumption of foods and beverages rich in polyphenols such as fruits, vegetables, cocoa, tea, and coffee, and the risk of all-cause mortality [3,4]. Despite the fact that it has been demonstrated that the consumption of 800 g/day of fruit and vegetables reduces the risk of cardiovascular disease and mortality from all causes, and 600 g/day reduces the risk of cancer, the WHO and several dietary guidelines still recommend 400 g/day [5]. However, most of the population does not even reach the minimum established by health authorities, which is equivalent to five portions of fruits and vegetables a day [6,7]. The benefits related to a healthy lifestyle are known worldwide, with new products emerging that offer the possibility of incorporating the aforementioned bioactive compounds through supplements/nutraceuticals, which helps to reach the nutritional recommendations [8].

In addition, due to the progressive aging of the population, among other factors such as diet and lifestyle, there is an accelerated increase in neurodegenerative diseases, with AD as the main exponent, whose prevalence doubles every 20 years. Due to the impact and social cost that it generates, many studies have been carried out on the relationship between the consumption of fruit and vegetables as a contribution of antioxidants that reduce oxidative damage at the brain level [9].

Polyphenols are produced as secondary metabolites of plants as a defense mechanism against adverse conditions (parasites, climate, or pollution, among others) [10] and form a broad group of bioactive phytochemicals, including phenolic acids, stilbenes, lignans, phenolic alcohols, and flavonoids [11,12]. They provide potential effects in the prevention and treatment of several pathologies, such as cardiovascular diseases, cancer, diabetes mellitus, and aging and neurodegenerative diseases [13] through several bioactivities (cardioprotective activity, anti-inflammatory activity, anti-aging activity, antioxidant activity, anticancer activity, antimicrobial effects, neuroprotective effects, lung protective effects, kidney protective effects, prevention of osteoporosis, and protection against UV irradiation) [14]. Polyphenols attenuate inflammation by suppressing pro-inflammatory transcription factors from interacting with the proteins involved in gene expression and cell signaling [15]. The vasodilator, vasoprotector, antithrombotic, antilipemic, anti-sclerotic, anti-inflammatory, and anti-apoptotic capacity of polyphenols is a consequence of their healthy properties [12], considering that polyphenols are the main source of antioxidants (10 times higher than vitamin C and 100 times higher than vitamin E and carotenoids) [16]. In fact, quercetin, myricetin, and kaempferol are flavanols, the flavonoids with the highest free radical neutralizing capacity, as their phenolic group structure is capable of capturing the unpaired electrons of ROS, generating neutralized reactive species [17]. However, the concentration of polyphenols in plasma is highly dependent on the chemical structure and dietary source, so long-term intake is advisable to obtain all of the related benefits [12].

Vitamin E is well known for its antioxidant capacity; high plasma concentrations decrease lipid peroxidation [18,19] reducing cardiovascular risk [20]. Low levels of vitamin B12 are often closely related to homocysteine, an amino acid associated with oxidative stress [21]. Elevated homocysteine levels are associated with an increase in cardiovascular disease [22], and vitamin B12 is necessary for brain activity and is therefore essential for cognition. In fact, there is a close relationship between this compound, vitamin B12, and cognitive impairment [23]. Coming back to inflammation, oxidized low-density lipoprotein (OxLDL) is related to the initiation and progression of atherogenesis, characterized by chronic inflammation and accumulation of lipids and foam cells in the arterial endothelium [24]. Furthermore, TNF-α, sTNFR1, and C-reactive protein (CRP) are elevated in chronic inflammatory diseases as well as in patients at cardiovascular risk [25,26]. The consumption of fruits and vegetables or supplements rich in polyphenols and thyroid function is contradictory, it has been seen how certain foods and/or polyphenols can favor T3 and T4 biosynthesis, maintaining or even reducing TSH values [27,28] whereas compounds such as quercetin inhibit the production of thyroid hormones [29], banana and orange have antithyroid effects [30] while melon, watermelon or mango stimulate the production of T3 and T4 [31]. In the instance of catecholamines, hormones such as cortisol and noradrenaline are released in stressful situations, and compounds such as anthocyanins are able to modulate these hormones [32]. Moreover, catecholamine metabolism causes high oxidative stress by promoting the formation of a large number of superoxide radicals that can cause damage to proteins and membrane lipids [33].

The aim of the current clinical trial was to evaluate whether the long-term consumption of a high variety of polyphenols through a fruit- and vegetable-based nutraceutical is able to improve the biomarkers related to inflammation and oxidative status in healthy subjects who do not consume the recommended amounts of fruits and vegetables. In addition, the healthy non-elderly population has been the subject of few intervention studies that relate fruit and vegetable consumption and catecholamine levels as variables related to cognition [27,28,34].

## 2. Results

### 2.1. Study Population

A total of 108 subjects were included in this study (Table 1 shows their demographic data), divided into two homogenous groups (N_1_ and N_2_). Finally, after cross-over and loss to follow-up, 92 subjects (N_1_ = 48, N_2_ = 44) completed the intervention and were included in the final statistical analysis, as depicted in Figure 1.

### 2.2. Plasmatic Biomarkers

The means and standard deviations of all of the biomarkers analyzed in all subjects, both at baseline and at the end of the intervention after the polyphenolic extract intake (EXT) or placebo intake (PLA), are depicted in Table 2.

In the EXT group, the parameters related to inflammation, such as OxLDL (Δ 9.46 ng/mL), sTNFR1 (Δ 0.13 ng/mL), and CRP (Δ 0.11 mg/L), showed a significant reduction (*p* < 0.05), which was not observed for the PLA group (*p* > 0.05). In the case of TNF-α, it decreased significantly in both the EXT and PLA groups (Δ 0.6 pg/mL and 0.39 pg/mL, respectively). Nevertheless, the reduction was more significant in the EXT group (*p* < 0.001) compared to the PLA group (*p* < 0.05). Comparing between groups at the end of the intervention, the results showed significant differences (*p* < 0.001) only in OxLDL, CRP and dopamine, as shown in Figure 2a,b and Figure 3, respectively; on the contrary, sTNFR1 (*p* = 0.089) and TNF-α (*p* = 0.09) did not reach significance.

Moreover, the cardiovascular risk can be determined by homocysteine. In fact, both groups showed a significant reduction in homocysteine, especially the EXT group, decreasing to Δ 1.32 mcm/L (*p* < 0.001) in contrast to the PLA group (Δ 0.68 mcm/L; *p* < 0.05). A comparison between groups at the end of the intervention showed non-significant differences (*p* = 0.053).

Regarding the lipid profile, the variation in high-density lipoprotein (HDL) cholesterol depending on the type of product consumed was remarkable. The HDL values reduced significantly after PLA consumption (Δ 4.52 mg/dL; *p* < 0.05), while increased significantly in the EXT group (Δ 6.25 mg/dL; *p* < 0.05). Additionally, the difference between both groups was also statistically significant at the end of the intervention (*p* < 0.05). Finally, total cholesterol and triglycerides values did not show significant differences throughout the intervention after PLA or EXT consumption.

Regarding the vitamins analyzed, a slight increase in vitamin B12 was observed in both groups, as well as small non-significant variations in vitamin E values, as shown in Table 2 (*p* > 0.05). For example, vitamin B12 varied after the intake treatment in both the PLA and EXT groups; however, significant differences were not found (*p* = 0.43 and *p* = 0.20, respectively). In the case of vitamin E, the values varied in the PLA (*p* = 0.98) and EXT (*p* = 0.91) groups. Moreover, comparing between groups at the end of the intervention, no significant differences were observed for either vitamin B12 (*p* = 0.205) or vitamin E (*p* = 0.974).

Concerning the catecholamine values, the main finding was the significant increase in dopamine (Δ 4.18 pg/mL; *p* < 0.001) after consumption of EXT with respect to PLA (*p* = 0.22), showing significant differences between groups at the end of the intervention (*p* < 0.001). A significant increase in norepinephrine values was observed (*p* < 0.05), with an increase of Δ 39.21 pg/mL in the EXT group, not observed in the PLA group (*p* = 0.72). There was no significant difference between the groups at the end of the study. Analyzing the cortisol and adrenaline values, no significant differences were observed in any group during the intervention, nor at the end of the intervention.

Thyroid function was determined by the hormones T3, T4, and TSH. During the intervention, it was observed that the T3 hormone values decreased significantly in both groups (*p* < 0.05). The T4 hormone values ranged in different ways, increasing significantly in the PLA group (Δ 0.57 mcg/L; *p* < 0.001) and decreasing non-significantly in the EXT group (Δ 0.47 mcg/L). Meanwhile, the TSH hormone values remained constant after the consumption of both products. Comparing the evolution of the three hormones between groups at the end of the study, no significant differences were observed for T3 (*p* = 0.195), T4 (*p* = 0.195), or TSH (*p* = 0654).

To determine the safety of the products under investigation, transaminases (GOT, GPT, and GAMMA GT) were analyzed in order to assess the liver function of the subjects, not observing significant changes (*p* > 0.05) in any of the aforementioned liver enzymes after consumption of either the product or the placebo. Therefore, it can be assumed that the intake of the products is safe.

## 3. Discussion

As commented above, the data obtained in the present manuscript reflect some differences between treatments. However, under basal conditions, no significant differences were observed between groups, which reinforces the significant changes obtained after the product consumption.

Plasmatic homocysteine is considered an indicator of cardiovascular risk that can be modified through exercise and diet. In addition, its high neuroprotective effect is well-known [35,36]. However, it seems that food provides a constant supply of nutrients to maintain optimal homocysteine metabolism [37]. In our study, homocysteine decreased significantly after the product consumption, as in previous studies [22,38]. Therefore, the consumption of polyphenols in high doses helps to reduce homocysteine, which could be considered as cardioprotective capacity. In turn, lipid profile, total cholesterol and triglycerides were not affected throughout the intervention. However, it was noteworthy that the plasmatic HDL cholesterol increased significantly after the consumption of the product and decreased after the consumption of the placebo. These changes in plasmatic lipoproteins reinforce the idea of the cardioprotective capacity of the product under investigation.

The main finding in the present research was the significant decrease in OxLDL, together with the increment in HDL cholesterol after the chronic consumption of the polyphenol-rich extract with respect to the placebo. These results are in concordance with previous studies supporting that the polyphenols show protection against atherosclerosis, thereby inhibiting vascular smooth muscle proliferation [12,39,40]. OxLDL is one of the few parameters recognized by the European Food Safety Authority (EFSA), in the framework of the function claims, and appears to be an appropriate outcome variable for the substantiation of health claims related to the reduction in oxidative damage to lipids [41,42]. Moreover, OxLDL promotes the adhesion of monocytes to endothelial cells through a mechanism independent of the expression of ICAM-1 and VCAM-1, generating foamy cells [43]. However, this does not occur with LDL [44]. This binding is performed through certain receptors, such as CD36. OxLDL is a ligand for PPARγ and upregulates the expression of CD36, favoring the uptake of OxLDL to macrophages [45] and consequently contributing to the development of atherosclerosis through immune system activation [46]. Interestingly, OxLDL promotes endothelial dysfunction and atherosclerosis, thus decreasing cardiovascular risk [47,48].

In turn, the treatment with the product under investigation led to plasmatic reduction of CRP, an acute-phase protein synthesized in the liver, mediated by other cytokines, especially IL-1 and IL-6 [49]. High levels of CRP are correlated with an increased risk of cardiovascular events, even in healthy individuals [50]. Several studies have found an association between CRP concentrations and cardiovascular events such as acute myocardial infarction, stroke, and the progression of peripheral arterial occlusive disease [51,52]. Although the levels obtained in CRP are within the normal ranges for a healthy person, the decrease in these levels shows the validity of polyphenols in this regard.

Moreover, the immune inflammatory marker TNF-α was significantly reduced after the consumption of the extract. TNF-α is a pro-inflammatory cytokine that is significantly expressed in adipose tissue, as well as in leukocytes, endothelial cells, and muscle cells [53], participating in endothelial dysregulation, stimulating monocyte and macrophage migration, and inducing the expression of adhesion molecules [54]. In fact, an increase in TNF-α can occur in a diet with a low concentration of carotenes, which can explain the reduction observed in the TNF-α levels [55]. Therefore, it makes sense that in our study it decreases its value after product consumption due to the high presence of carotenes in the product.

The polyphenolic extract used for the present investigation has been tested in several intervention studies, with different consumption times and in different population groups, leading to inconsistent results. In a 28-week intervention in trained men belonging to the Austrian special police forces, significant reductions (*p* < 0.001) were reported in the concentration of carbonyl groups in proteins and TNF-α in the 16th and 28th weeks. A difference with our study was that in their study samples were obtained at weeks 4, 8, 16, and 28. Notably, the values increased up to week 8 and decreased from week 16 onwards [56]. In an eight-week intervention in overweight and obese subjects over 40 years old, significant reductions in total cholesterol, LDL cholesterol, and TNF-α were reported, which were not obtained with the placebo; nevertheless, no significant differences were found in CRP, sTNFR1, or OxLDL [57]. Both results seem to confirm that long-term polyphenol intake, as in our study (16 weeks), is associated with greater effectiveness against inflammation. In another eight-week intervention in obese women, significant improvements were observed in several protein oxidation markers such as carbonyl proteins, OxLDL (*p* = 0.015), and total lipid oxidation of all lipids. Regarding inflammatory markers, significant reductions in TNF-α (*p* = 0.011) were noticed. At baseline, the TNF-α values were above the established range in the placebo and extract groups; at the end of the intervention, they remained elevated in the placebo group, and decreased to normal values in the group that consumed the product [58]. Finally, the effectiveness of the extract has also been tested in non-healthy subjects, reporting a reduction in OxLDL after a three-month consumption period by heavy smokers [59].

Regarding catecholamines, significant increases were observed in norepinephrine and, especially, dopamine after consumption of the product under investigation. However, no significant changes were observed for cortisol and adrenaline in any group. Despite some previous studies having reported a reduction in cortisol in plasma [60] or saliva [61] after polyphenol intake, this is not usual, and the values rarely show significant changes [62,63,64] regarding polyphenol intake. The fact that both the adrenaline and cortisol values did not increase implies that an adrenergic response linked to the consumption of the product did not develop. Noradrenaline and dopamine are also increased in elderly patients with neurodegenerative diseases [65,66], and are related to anxiety [67] and cognitive enhancement [68]. Moreover, norepinephrine is closely related to attention, as well as alertness [69,70]. The results obtained in the present study in terms of dopamine and noradrenaline could be related to the improvements at the cognitive level observed in another publication [71]. We can affirm that the consumption of polyphenols therefore increases the levels of alertness and attention, which correlate with results at the level of cognition tests [71].

A limitation of the present clinical trial, as described by Di Lorenzo et al. [10] in a systematic study on polyphenol bioavailability, is that the inclusion of healthy subjects in clinical trials makes it difficult to assess significant differences in biomarkers, which are altered under pathological conditions. Other limitations of this study include the fact that the metabolites of phenolic compounds in the product were not determined beyond the vitamin E content or individual response, which could affect polyphenol bioavailability and bioactivity [72].

It is important to remember that the volunteers of the present study are healthy subjects and the inflammation biomarkers that we have determined are between normal values in plasma. Therefore, the differences observed within the range of normal values for these biomarkers, despite being statistically significant, are not striking from a clinical point of view. However, if the product under study has achieved statistically significant changes in healthy people, these changes could be even greater in patients with abnormal values [73,74], since the polyphenols in the product would help combat at least partially- the agent causing this disruption of plasma level [75].

Therefore, future research could focus on considering these aspects in both healthy and non-healthy population.

## 4. Materials and Methods

### 4.1. Trial Design

This study consisted of a randomized, cross-over, double-blind, sex-stratified, and placebo-controlled clinical trial to assess the effectiveness of daily consumption of an encapsulated nutraceutical (Juice Plus+ Premium^®^, The Juice Plus+ Company, Collierville, TN, USA) based on different fruits and vegetables on blood plasma biomarkers of inflammation and oxidative status. The intervention had a length of 36 weeks—two periods of 16 weeks separated by a washout period of 4 weeks. During the intervention, the subjects came to the laboratory four times (at the beginning and end of each phase), obtaining blood samples under fasting conditions. At the beginning of each phase, the product (EXT or PLA) was delivered to the subjects, and at the end of each phase, the subjects were required to return the packages for the researchers to quantify the remaining products.

Each subject signed an informed consent document and was assigned a numerical code in order of arrival; subsequently, an outside researcher carried out the randomization using a computed generator (Epidat v4.1 Epi-dat, Spain) assigning the subjects to one of the groups. Neither the researchers nor the subjects knew which groups the subjects belonged to. Both the product and the placebo were marked with codes provided by the distributed company and classified into A and B, and only at the end of the study were the researchers notified of which code corresponded to the product and which to the placebo. The protocol was approved by the Institutional Review Committee of the Catholic University San Antonio of Murcia (UCAM) (date: 24 November 2017; code: CE111072). This study was carried out following the Standards of Good Clinical Practice and was conducted according to the Declaration of Helsinki. The trial was registered at www.clinicaltrials.gov (accessed on 11 June 2020) (identifier CFE/JU/44-17.) The study was carried out in the Pharmacy Department of the Faculty of Health Sciences of the Catholic University San Antonio of Murcia (UCAM). Current European legislation on the protection of personal data was complied with (Regulation (EU)2016/679).

### 4.2. Participants

The subjects had to fulfill all of the criteria of inclusion (sign the informed consent form, BMI between 18.5 and 35 kg/m^2^, not have a chronic disease, not consume more than three servings of fruit and vegetable per day, and aged between 18 and 65 years) and not meet any of the exclusion criteria (being on medication or undergoing pharmacological treatment, taking multivitamins, allergic to fruit or vegetables, being on a diet, being vegetarian or vegan, smoking, consuming more than 3 glasses of alcohol (wine or beer) a day, being pregnant, had major surgery in the last three months, having sleep problems, and having donated 0.5 L of blood in the last three months). After checking the inclusion/exclusion criteria, a total of 108 subjects of both sexes started the study. Verification was undertaken to ensure that the subjects continued to meet the criteria throughout all of visits to the unit by means of a Mediterranean lifestyle index (MEDLIFE) survey to assess dietary habits and physical activity [76]. During the intervention, 16 subjects were lost to follow-up, so the number of subjects who completed the study was 92, whose demographics data are shown in Table 1.

### 4.3. Test Supplement

The product under investigation and the placebo had the same characteristics, and both were manufactured and provided by The Juice Plus+ Company, Collierville, TN, USA. The presentation was in the form of white encapsulated pharmaceutical products, provided in bottles differentiated only by a code. Once the study was completed, the investigators were provided with which code belonged to the product and which to the placebo. The differences in the commercial product were that it consisted of an intake of three different types of capsules per day—two of mixed berries, two of mixed fruits, and two of mixed vegetables, each provided in a separate bottle. For this reason, and in order to be able to manufacture a similar product and placebo, it was decided to manufacture capsules containing a mixture of the three aforementioned capsules (berries, fruit, and vegetables), whereby the daily consumption for this study was six capsules—three before breakfast and three in the mid-afternoon, accompanied by water.

The product under investigation was provided in opaque white capsules and contained fruit, vegetable and berry juice powder and pulp from apple, orange, pineapple, cranberry, peach, acerola cherry, beet, prune, date, mango, carrot, parsley, kale, broccoli, cabbage, spinach, rice bran, tomato, artichoke leaves, bilberries, blackberries, black currant, blueberries, cranberries, concord grapes, elderberries, grape seed extract, green tea extract, ginger root, raspberries, cocoa, pomegranate, natural tocopherol blend (γ, δ, α, β-tocopherols, sunflower α-tocopherol), natural carotenoid blend (lutein, β-carotene, lycopene, astaxanthin), citrus pectin, citrus bioflavonoids, lemon peel, calcium carbonate, garlic powder, spirulina, natural enzyme blend, silicon dioxide, vegetable derived magnesium stearate, tangeretin and Lactobacillus acidophilus. A daily dose of 6 capsules provided 2.91 mg β-carotene, 18.7 mg vitamin E, 159 mg vitamin C, 318 μg folate, 6.1 mg lutein, 1 mg lycopene and 0.15 mg astaxanthin. The placebo consisted of microcrystalline cellulose, rice starch, vegetarian capsule (cellulose), magnesium stearate and artificial colours (FD&C yellow #6 and FD&C blue #1).

The polyphenolic characterization of the product was performed in a previous study by Bresciani et al. through UHPLC-QqQ-MS, showing a total of 119 polyphenolic compounds of different phenolic families, including flavanols such as kaempferol and quercetin, anthocyanins, and flavones [77]. Regarding bioavailability, knowledge of the bioavailable quantity of polyphenols is more interesting than the total quantity, since, in many situations, the most abundant polyphenols are not the most active for the organism [78]. For this reason, Bresciani et al. evaluated the bioavailability of the product, finding that of the 92 molecules monitored, only 20 circulating metabolites were identified in plasma, all of them as remains of sulfate, glucuronide, or glycine, appearing at different times in plasma depending on the absorption location in the digestive tract [79].

### 4.4. Study Variables

All of the variables were analyzed four times during the study: At the beginning and at the end of each 16-week product consumption phase. During the study, the subjects were not allowed to begin or modify any hormonal treatment, nor make any significant dietary or physical activity changes that could affect the study variables.

#### Blood Sample Measurements

The subjects had to arrive after a 12 h fasting period, only allowing water intake in the previous 3 h. Moderate–high-intensity exercise was not allowed in the 24 h prior to extraction. Blood samples were collected from the antecubital vein in a k3E/K3EDTA blood collection tube (3 mL) and Serum Gel and Clot Activator blood collection tubes (5 mL) *3. Before being sent to the laboratory (Laboratorios MUNUERA S.L. Murcia, Spain), only Serum Gel and Clot Activator blood collection tubes were centrifugated (4500 rpm, 5 min, 4 °C). The biomarkers analyzed for each variable were: (a) Pro-inflammatory biomarkers: OxLDL (ng/mL) by ELISA assay in serum samples, TNF-α (pg/mL) and sNTFR1 (ng/mL) by competitive ELISA (DRG diagnostic brand), CRP (mg/L) by turbidimetric immunoassay (PETIA), and homocysteine (pg/mL) on a BN ProSpec^®^ analyzer (according to the protocol provided in the Siemens N Latex HCYOPAX 03 kit); (b) vitamins: Vitamin E (mcg/mL) by HPLC with a JASCO chromatograph (PU-980 Intelligent PLC-Plus) and a Betasil-C18 column (particle size 5 µm, 250 × 4.6 mm) connected to a JASCO fluorescence detector (FP-920 Intelligent Fluorescense Detector), and vitamin B12 (pg/mL) by chemiluminescence with commercial immunoassays using Centaur XP Siemens Diagnostics automated equipment (Siemens Healthcare S.A.); (c) catecholamines: Cortisol (mcg/L) using the chemuminescence technique with Immulite 2000 immunoassay (DPC, Gwynedd, R.U.), and noradrenaline (pg/mL), adrenaline (pg/mL), and dopamine (pg/mL) by HPLC with a flow of 0.5mL/min, Tº of 35 °C, sensitivity of 20, potential of 0.7, mobile phase of acetonitrile–water, and reverse phase of C18 using n electrochemical detector; (d) thyroid hormones: T3 (mcg/L), T4 (mcg/L), and TSH (mU/l) using Centaur XP Siemens Diagnostics automated equipment (Siemens Healthcare S.A.); (e) immune status: IgA (mg/L) using the turbidimetry technique with the Monolab test. As reagents, Tris buffer (20 mmol/L), PEG 8000, and pH 8.3 were used as the diluent and goat serum, anti-human IgA, and pH 7.5 as the antibody.

### 4.5. Statistical Analysis

The continuous variables are presented as mean ± standard deviation (SD) at baseline and in their evolution, applying the Kolmogorov–Smirnov test to verify the normal distribution of the discontinuous data. The analysis was performed for all of the subjects who participated in the study, in the period of both product and placebo consumption. For the quantitative variables, a Student’s *t*-test comparison was performed between the two branches of the study. To analyze the differences between the groups (EXT and PLA) in the evolution of the different variables, a repeated measures analysis of variance was performed with time as the intrasubject factor. In this way, differences were established for each variable analyzed, considering these factors. A Bonferroni test was carried out for the post-hoc analysis. Significant differences were compared (with the option of assuming equality of variance or not). In the set of statistical tests, the significance level chosen was 0.05. Statistical analysis was carried out with SPSS 24 software (SPSS, Inc., Chicago, IL, USA).

## 5. Conclusions

Our research showed that the chronic consumption of a polyphenolic extract based on berries, fruits, and vegetables in a healthy population that does not eat the recommended amounts of fruits and vegetables per day improves oxidative status and inflammation through an improvement in several blood biomarkers, especially OxLDL, CRP, and HDL cholesterol, also improving the values of catecholamines—particularly dopamine.

## Figures and Tables

**Figure 1 molecules-26-03604-f001:**
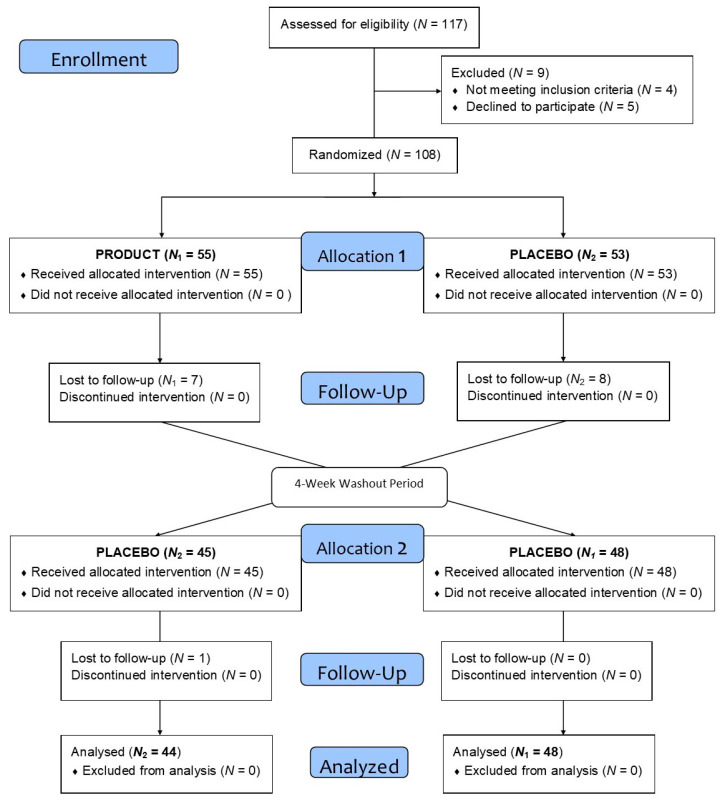
Flow chart of the present study.

**Figure 2 molecules-26-03604-f002:**
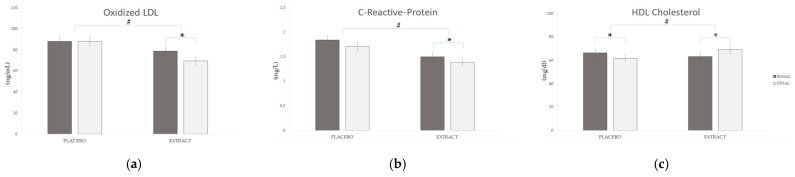
Evolution of (**a**) oxidized low-density lipoprotein (OxLDL), (**b**) C-reactive protein (CRP), and (**c**) high-density lipoprotein (HDL) during the study. * Means significant statistical differences when comparing the evolution between the baseline and final (*p* < 0.05). # Means significant statistical differences when comparing the evolution between groups (*p* < 0.001).

**Figure 3 molecules-26-03604-f003:**
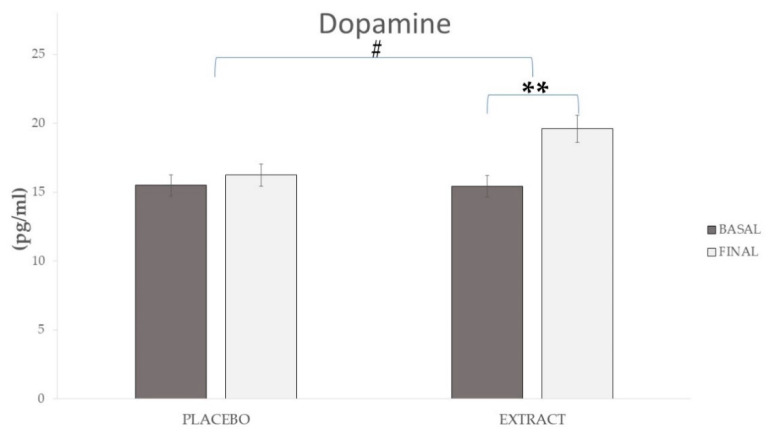
Evolution of dopamine during the study. ** Means significant statistical differences comparing the evolution between baseline and final (*p* < 0.001). # Means significant statistical differences comparing the evolution between groups (*p* < 0.001).

**Table 1 molecules-26-03604-t001:** Demographic data of the volunteers in the present study.

	Total	N_1_	N_2_
*N*	92	48	44
Men	45 (48.91%)	20 (44.44%)	25 (55.56%)
Women	47 (51.09%)	28 (59.57%)	19 (40.43%)
Age (years)	34 ± 11	33 ± 10	36 ± 12
Weight (kg)	73.10 ± 14.29	70.68 ± 13.88	75.68 ± 14.44
Height (m)	1.72 ± 9	1.71 ± 9	1.73 ± 9
BMI (kg/m^2^)	24.40 ± 3.43	23.87 ± 3.42	24.99 ± 3.38

**Table 2 molecules-26-03604-t002:** Evolution of the different biomarkers measured during the study. Values are expressed as mean and standard deviation at baseline and at the end of the intervention. *p*-Value corresponds to the comparison between products at the end of the intervention.

		Baseline	Final	*p*-Value
TNF-α (pg/mL)	Placebo	5.69 ± 1.02	5.30 ± 0.87 *	0.609
	Extract	5.82 ± 1.14	5.22 ± 1.21 **	
sTNFR1 (ng/mL)	Placebo	1.43 ± 0.32	1.43 ± 0.33	0.089
	Extract	1.49 ± 0.37	1.36 ± 0.26 **	
Homocysteine (mcm/L)	Placebo	11.75 ± 2.21	11.07 ± 2.92 *	0.053
	Extract	11.65 ± 2.3	10.33 ± 2.27 **	
Vitamin B12 (pg/mL)	Placebo	309.11 ± 85.51	314.99 ± 82.20	0.205
	Extract	308.41 ± 123.51	333.45 ± 115.00	
Vitamin E (mcg/mL)	Placebo	13.20 ± 3.33	13.21 ± 3.53	0.974
	Extract	13.15 ± 3.68	13.20 ± 3.37	
Cortisol (mcg/L)	Placebo	14.32 ± 3.22	13.92 ± 3.72	0.93
	Extract	13.42 ± 2.07	13.87 ± 2.03	
Adrenaline (pg/mL)	Placebo	22.05 ± 8.03	23.50 ± 6.98	0.7
	Extract	22.67 ± 8.61	23.91 ± 6.97	
Norepinephrine (pg/mL)	Placebo	361.86 ± 50.58	367.06 ± 52.28	0.322
	Extract	343.64 ± 46.38	382.85 ± 43.88 **	
T3 (mcg/L)	Placebo	1.13 ± 0.23	1.07 ± 0.18 **	0.195
	Extract	1.1 ± 0.23	1.03 ± 0.16 **	
T4 (mcg/L)	Placebo	7.64 ± 1.31	8.21 ± 1.31 **	0.781
	Extract	8.74 ± 0.75	8.27 ± 1.03	
TSH (mU/L)	Placebo	2.28 ± 0.51	2.27 ± 0.53	0.654
	Extract	2.23 ± 0.61	2.20 ± 0.51	
Total cholesterol (mg/dL)	Placebo	184.71 ± 40.21	188.78 ± 39.07	0.773
	Extract	184.06 ± 36.30	187.13 ± 32.07	
Triglycerides (mg/dL)	Placebo	74.07 ± 26.92	73.94 ± 29.27	0.178
	Extract	69.79 ± 24.93	68.39 ± 21.54	

* Means significant statistical differences when comparing the evolution between the baseline and final (*p* < 0.05). ** Means significant statistical differences when comparing the evolution between the baseline and final (*p* < 0.01).

## Data Availability

Not necessary.

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
