# Peer review of "Effects of a Fruit and Vegetable-Based Nutraceutical on Biomarkers of Inflammation and Oxidative Status in the Plasma of a Healthy Population: A Placebo-Controlled, Double-Blind, and Randomized Clinical Trial"

_molecules, 2021, doi:10.3390/molecules26123604_

Round 1

Reviewer 1 Report

In general de manuscript shows the results from a placebo controlled cross-over intervention study with a food supplement.

The results are interesting however the statistical analysis used might not be the best given rise to results that show significance for different parameters even with the placebo treatment. Additionally it is unusual to find statistical significance also in the baseline levels for some biomarkers (oxLDL or CRP for instance) between treatments. If the authors think that this is a real effect or that the washout period has not be enough they should comment on it. However the impression is that the statistical model used is creating this fake effect.

The abstract should be improved. There is no point to write in the abstract that the composition of the nutraceutical was previously determined. Please give some detail on the composition (mg polyphenols or vitamin c or some kind of information that can give an idea of the amount of fruit and vegetables contained in the supplement). Also, there is no point of including the term cancerous in the abstract when there is no reference to such activity in the manuscript. Inflammation and redox state are sufficient in this sense.

The introduction can be improved. In general it is important to highlight the importance of consuming fruits and vegetables and the fact that most studies have shown that the use of substitutes in general has lower effect or no effect at all. The use of supplements can be really useful to asure this kind of studies completion by the volunters and it is justified in this case but it cannot justify the recommendation for this kind of substitutes for general population. This needs to be clarified.

In the results it is important to show that there are changes between treatments (or even without treatment at the two baseline points) when they are statistically significants, otherwise they cannot be considered real changes and so there is no point on showing them.

The changes in catecholamines need to be explained or the conditions for the assay better stadandarised. Catecholamines have a great fluctuation depending in many factors and here this fact is not considered. Additionally if the authors think that catecholamine results are important and need to be included they should explain what is the meaning of those results and the physiological relevance.

How do the authors explain the decrease in HDL values with the placebo?

Might have been an increase or decrease on fruit and vegetables or the diet in general during the intervention? has this been controlled somehow?

Reviewer 2 Report

Thank-you for conducting a useful piece of research that could be of value to readers. I would like to see more effort put towards drafting the introduction to better frame why the research was conducted, what is the proposed mechanism behind the nutraceutical and key nutrients in the product.

Ln 49 plats

Ln 62 tant

The key finding is the elevated dopamine. Can the authors tease this out in the discussion? What other studies have been done in this area? This seems novel, compared with the inflammatory markers, which really has been done a lot.

Why did the authors chose to measure the stress response, thyroid hormones and liver enzymes? I think these are valuable, but poorly justified in the introduction. Can the liver function tests be included in the study? I am confident the readers would like to see those results.

Finally, the study is funded by the supplement company. I have no issue with industry funded research, but the reader must understand the nature of the funding and the degree to which the company had input into the manuscript. Was there any embargo on publishing? Please define this relationship.

Reviewer 3 Report

The manuscript “Effects of Fruit and Vegetable-Based Nutraceutical on biomarkers of inflammation and oxidative status in plasma in Healthy Population: Placebo-Controlled, Double-Blind, and Randomized Clinical Trial” reports the results of an investigation aimed at exploring the effect of long-term consumption of a high-polyphenolic nutraceutical on different markers.

The manuscript provides interesting results, however an extensive revision is needed to improve clarity and the ease of reading.

MAJOR COMMENTS

  • First of all, the manuscript is generally not very written, and an English editing is needed, preferably by an English native speaker
  • It is not clear the rationale behind the study and the hypothesis authors aimed to explore. Moreover, it might be helpful to clarify the hypothesis behind the choice of the plasma marker, as they space in very different tasks, i.e. thyroid, vitamin B12, etc.
  • Introduction should be revised to better elucidate the rationale that led to the study
  • One of the main limits of the study is the choice to test the biological activity of the nutraceutical on healthy subjects (they fall in the range of a normal BMI). In fact, authors find for example differences in the values of CRP or HDL Cholesterol after the treatment, but these values falls within physiological ranges (es. CRP: from 1.50 mg/L to 1.39 mg/L , both the values fall in a physiological range), and the discussion at lines 205-209 becomes not so useful. So, what’s the rationale behind these values? I think that authors should strive to describe not only the statistical differences but also the clinic relevance of these changes and there is a miss of limitation acknowledgement in discussion session about this.
  • Another limitation of the study is the fact that authors did not really proof that the intake of the nutraceutical pill allowed the presence in the bloodstream of the compounds present in the formulation. Despite authors claim as proof the paper at ref. 62, the inter-individual variability of the compounds in different populations is nowadays clear. So, it would have been helpful to have a biomarker of intake of such phenolic compounds. This is clearer for vitamin, for example. In the formulation there is an amount of vitamin E, but the plasma vitamin E has not been affected by this formulation, so it is tricky to address effects to the formulation. Linked to this, which is the meaning of the assay of vitamin B12, which is a vitamin contained in animal-based products?
  • The authors did not provide any data bout the diet followed during the intervention. Did the subjects follow a specific diet or a free-living diet with no restriction? Could they consume fruit and vegetables in addition to the test products? It is well known that several foods may affect the biomarkers under study, so authors should better specify the protocol and, if available, explain if the dietary intake of the different food group and the related nutrient intake differed between test product and placebo interventions
  • Figure 2 and 3 are just a repetition of the data present in table 2.
  • Bibliography can be expanded with several pertinent articles that have not been considered. Just some examples: doi:  10.1080/09637486.2019.1571021; https://doi.org/10.3390/foods9111606; DOI: 1016/S0140-6736(17)32253-5

OTHER COMMENTS

  • Line 15: despite composition was previously determined, it would be useful to add the amount of polyphenols provided by this product
  • Please take care of the measure units (e.g. for the height the unit measure is meter, m) and of the significant digits (age does not require any decimal digit, and I am sure that authors did not measure height at the hundredth of a cm).
  • Figure 2 is very small and difficult to read. The quality of the figure should be improved
  • Line 196: it should be specified that OxLDL, in the framework of the function claims, appears to be an appropriate outcome variable for the substantiation of health claims related to the reduction of oxidative damage to lipids, but not non disease risk reduction claims (http://dx.doi.org/10.1016/j.numecd.2017.01.008)

Round 2

Reviewer 1 Report

Inflammatorybiomarkers, space between words

The sentence form 55

a broad group of bioactive phytochemicals, highlighting; phenolic acids, stilbenes, 56

lignans, phenolic alcohols and flavonoids” needs to be rephrased. The same for “Provided potential effects in the prevention and treatment of several pathologies, such as cardiovascular diseases, cancer, diabetes mellitus, aging and neurodegenerative diseases” maybe they provide…

quercetin, myricetin 68 and kaempferol are flavonols (also flavonoids, but they are all three flavonols and this could be said).

“due to phenolic group structure is capable of capturing the unpaired electrons” please rephrase

However, the concentration of polyphenols in 71

plasma is highly dependent on the chemical structure and dietary source, so it is advisable 72

long-term intake to reach all related benefits [12].this sentence makes no sense

In the other hand for on the other hand…anyway this sentence, again, makes no sense

There are still no mention or justification for the inclusion of catecholamines in the introduction

In general, the discussion should be improved because it is full of assumptions that are not real and in no case are related to the results of the present investigation. For instance: “We could affirm that the consumption of polyphenols

therefore increases the levels of alertness and attention and we could correlate it with results at the level of cognition tests.” Therefore? this are not results from your own work. Maybe if you think that polyphenols intake increases epinephrine and that this change can be considered statistically significant it may explain the results encountered by other authors on attention. This is said before anyway.

Likewise, neither for the 2 vitamins evaluated (vitamin E and vitamin B12) and thy- 211

roid function (T3, T4, TSH) no significant changes were observed during the intervention

Likewise? There is no need to comment on this. In fact, the reason why you decided to measure thyroid function is not clear.

You should first explain in the introduction or in the methodology why you are measuring the things you have measured.

What is the meaning of “compounds present int polyphenols”, polyphenols are the compounds, and are present in foods or supplements? Not sure what you mean.

Later “validity of the compound in this regard” which compound? the food supplement?

“the increase of TNF-α can occurs in diet with low concentration of carotenes”??

,

The results in the paragraph starting with “

Were obtained by using the same supplement as in the present work?

“the 16th and 28th weeks” for on the 16th

decreased form should be from

systematic revies

difficulty the assessment

Future Directions Therefore, future research could focus studies taking these aspects

into account.

The discussion should be largely improved. It is full of inconsistences and mistakes. There is not a line of discussion. Are all supplements the same in all papers? Are they characterized by the presence of the same fruits or vegetables are the ones in the present work?

These criteria were verified 314

that they continued to be met throughout all visits to the unit by means of a survey. Dur- 315

ing

Which kind of survey was used? This should be included in the methodology. If there was no survey it is ok but it should be said and it should be comment what effect could have in the study the fact that volunteers changed their dietary habits during the 16 weeks of intervention.

Ref 21? Santos, M.C. de S. dos;?

Refe 22, it looks like there is an extra space

Reviewer 2 Report

The manuscript has been improved with regards to justification and background in the introduction, however the English language editing is still required. In it's current form, it is not appropriate for publication. The authors have been asked by numerous reviewers to redraft the manuscript for grammar.

The authors have also been asked to add full financial arrangement of the funding supplied by Juice Plus. The manuscript as it currently stands appears to be completely commercially driven. Please add into the manuscript a full disclosure of the funding arrangement. The manuscript should not be published in it's current form.

Finally, other reviewers have asked the authors to address the issue of clinical versus statistical significance in the interpretation of CRP changes and other inflammatory changes that appear statistically significant, but really appear to be clinically irrelevant. Some effort on behalf of the authors to adequately address this is required.

Reviewer 3 Report

I thank the authors for addressing most of the comments from the past revision.

English still needs to be carefully revised and the manuscript would benefit of a deep revision because several sentences (e.g. in the discussion section)  are not in a logical sequence making the manuscript not easy to read.

More, I have some others minor comments to be considered:

Lines 87: provide references for such statement

Lines 88-89: the link with previous sentence is not clear. Please rephrase

Tab 1: two significant digits for age, height (in cm) have no sense. Please revise

Lines 174-177: sentence not clear
